**communications** engineering

# Reinforced optical cage systems enable drift-free single-molecule localization microscopy

Hao Qiu[1,2,6], Matthew C. Tang[1,3,6], Selene K. Roberts [1], Guoli Li[4], Rong Su[4], Marisa L. Martin-Fernandez[1], David T. Clarke[1], Shugang Liu[5], Xiaojie Liu[2] & Lin Wang [1] ✉

Single-molecule localization microscopy achieves nanometer-scale resolution but is compromised by sample drift during image acquisition. Here we present reinforced optical cage systems, a novel approach that eliminates drift at its mechanical source rather than correcting it through complex image post-processing or fiducial markers. Reinforced optical cage systems employ perforated optomechanical components interconnected by tungsten-steel rods in a design proven by mechanical stability simulations. Our bench-top microscope, built with reinforced optical cage systems, demonstrated exceptional three-dimensional stability, with mean cumulative lateral drift of approximately 5 nanometers over 2 h in widefield fluorescence microscopy and 11-16 nanometers over 15 min in single-molecule localization microscopy, free from measurable axial drift. This development allows super-resolution microscopy to reach its full resolution without the necessity of sample drift correction, offering a straightforward, cost-effective, low-maintenance, and readily accessible solution to high-performance super-resolution microscopy. By addressing the fundamental issue of mechanical instability, reinforced optical cage systems enable improved precision instrumentation for the broader scientific and engineering community.

The advancement of single-molecule localization microscopy (SMLM) has enabled the application of light microscopy at resolutions surpassing the diffraction limit. The main principle of SMLM is the temporal separation of spatially overlapping point spread functions (PSFs) of single molecules, such that the centroids of each molecule can be localized with high precision following the fitting of a Gaussian function to each PSF[1]. Accumulating the localizations of the molecules then forms a super-resolution image. The SMLM family consists of several methods, such as photoactivation localization microscopy (PALM)[2], stochastic optical reconstruction microscopy (STORM)[3], and DNA point accumulation for imaging in nanoscale topography (DNA-PAINT) microscopy[4]. In PALM, photoactivatable fluorescent proteins are photoinduced into a fluorescent state, while in STORM, organic dyes are photoswitched using high-power laser illumination and an appropriate buffer solution. Finally, DNA-PAINT exploits the transient association between pairs of short complementary DNA sequences with one oligonucleotide conjugated to a target-specific probe and the other conjugated to a dye molecule. Fast binding and unbinding generate a sequential single-molecule signal that can be localized using the same strategy in PALM and STORM.

A key commonality across these methods is that sufficient localizations must be collected to reconstruct an accurate super-resolution image, necessitating long image acquisition times, typically about 15–30 min[1]. Combined with the high laser power (>1 kW/cm²) used in STORM, samples often drift from their initial locations along lateral (x and y) and axial (z) directions due to the mechanical instability of microscopes[5] and temperature changes[6], both of which degrade the resolution and fidelity of super-resolution images[7]. Sample drift also degrades image quality in other super-resolution microscopy techniques, such as super-resolution radial fluctuations (SRRF) microscopy[8,9], which require a large volume of raw images for super-resolution image reconstruction.

Various methods exist to mitigate sample drift in SMLM. Axial drift can be addressed through active stabilization of the focal plane using a piezoelectric-driven objective lens[10,11]. By capturing images continuously and estimating the axial position of the focal plane, the axial position of the

[1]Central Laser Facility, Research Complex at Harwell, Rutherford Appleton Laboratory, Science and Technology Facilities Council, Didcot, UK. [2]School of Electrical and Information Engineering, Jiangsu University of Technology, Changzhou, Jiangsu, China. [3]Division of Structural Biology, Nuffield Department of Medicine, University of Oxford, Oxford, UK. [4]Shanghai Institute of Optics and Fine Mechanics, Chinese Academy of Sciences, Shanghai, China. [5]RayCage (Zhenjiang) Photoelectric Technology Co., Ltd, Zhenjiang, Jiangsu, China. [6]These authors contributed equally: Hao Qiu, Matthew C. Tang. ✉e-mail: lin.wang@stfc.ac.uk

objective lens can be adjusted to ensure the sample remains in focus. To address lateral sample drift, fiducial markers, such as gold nanoparticles[12], fluorescent beads[13], quantum dots[14], or micro-patterns[15], are commonly added around the samples. This approach leverages the persistent fluorescent emission observed in fluorescence imaging or the scattering inherent in bright-field imaging of fiducial markers, thereby facilitating the quantification of drift. Consequently, the drifted positions of localized molecules can be corrected in real-time during image acquisition[16] or by drift correction during image post-processing[17]. However, this approach requires additional steps of sample preparation, and careful regulation of the spatial distribution and density of fiducial markers is essential to avoid undermining the signal-to-noise ratio (SNR) due to the elevated background fluorescence[18]. Moreover, this approach is excessively dependent on fixed fiducial markers; even minimal vibrations of the fiducial markers immersed in solution may jeopardize the drift correction[19].

Another methodology to address lateral sample drift in SMLM involves data-driven statistical registration for drift detection and correction[20], such as redundant cross-correlation algorithms[21,22]. Typical workflows implement cross-correlation drift correction by splitting the image sequence into batches and comparing the locations of molecules in each batch to the first batch or to all other batches to estimate and correct drift. These methods do not require fiducial markers, reducing the complexity of sample preparation. However, the extent to which cross-correlation can be biased by the choices of parameters in correction algorithms is poorly understood[23]. Further, cross-correlation algorithms operate under the assumption that sample drift occurs progressively; consequently, they may be inadequate in instances where sudden and large drift is exhibited.

Alternative methods that do not require additional sample preparation or data-driven statistical image post-processing can correct sample drift by analyzing the speckle patterns of backscattered laser light in cells. While this approach reduces drift, it also introduces speckle-related imaging artifacts[24]. Another drift correction technique utilizes the displacement correlation analysis of bright-field image features of specimens illuminated with oblique light[25]. However, these methods require additional hardware and image processing.

All sample drift correction methods described thus far require either additional hardware, sample preparation, or image post-processing, thereby increasing experimental complexity. In this paper, we present a novel method for achieving SMLM with negligible sample drift using reinforced optical cage systems (ROCS). Evolving from innovative mechanical system designs, ROCS integrate each optomechanical component into a reinforced structure interconnected by tungsten-steel rods. A new standardized series of perforated cage components enables the formation of interchangeable ROCS modules. An SMLM system based on ROCS (ROCS-SMLM), as exemplified by ROCS-STORM, eliminates three-dimensional sample drift through an ultra-stable sample stage, obviating the need for additional hardware, sample preparation, or image post-processing. Our theoretical analysis and experimental validation demonstrate that residual sample drift in ROCS-SMLM is negligible and does not adversely affect image resolution. This approach offers a straightforward, inexpensive, reproducible, and low-maintenance solution, making high-performance super-resolution microscopies more accessible for biological imaging and beyond.

## Results
### Vibration modal analysis
Conventional optical cage systems suffer from inadequate mechanical stability, limiting their application in high-precision instrumentation. To overcome this limitation, we developed ROCS, inspired by the reinforced concrete structures used in civil engineering, where the reinforced bars are deployed to resist the shear, tensile, and compressive stresses[26]. ROCS utilize rigid tungsten steel rods to support and interconnect newly designed, perforated optomechanical components, resulting in highly stable optical systems.

The ROCS incorporate many novel perforated optomechanical components (see product catalog: https://www.raycage.com/sy). For example, the right-angle kinematic mirror mounts feature perforations, allowing four tungsten-steel rods to pass through in three orthogonal directions. These rods, analogous to reinforcing bars in civil engineering, support the mirror mount and rigidly interconnect it with cage plates, preventing mechanical drift of the mirror or dichroic mirror mounted inside (Fig. 1a). This ROCS assembly contrasts with conventional optical cage systems, where non-perforated mirror mounts are supported by a single post, and the stainless-steel rods provide neither structural support nor integrated connectivity (Fig. 1b).

Using Finite Element Analysis (FEA), we conducted vibration modal analysis on the right-angle kinematic mirror mount assembly using the ROCS as depicted in Fig. 1a. Modal analysis identifies the natural frequencies and mode shapes of a structure, which are determined by the stiffness of the system and mass distribution. In rigid-body dynamics, the first six modes of interconnected systems under the action of external forces correspond to the six degrees of freedom in space. By imposing constraints, as devised in the ROCS, these rigid-body displacement modes with a frequency of 0 can be eliminated, resulting in enhanced mechanical stability[27].

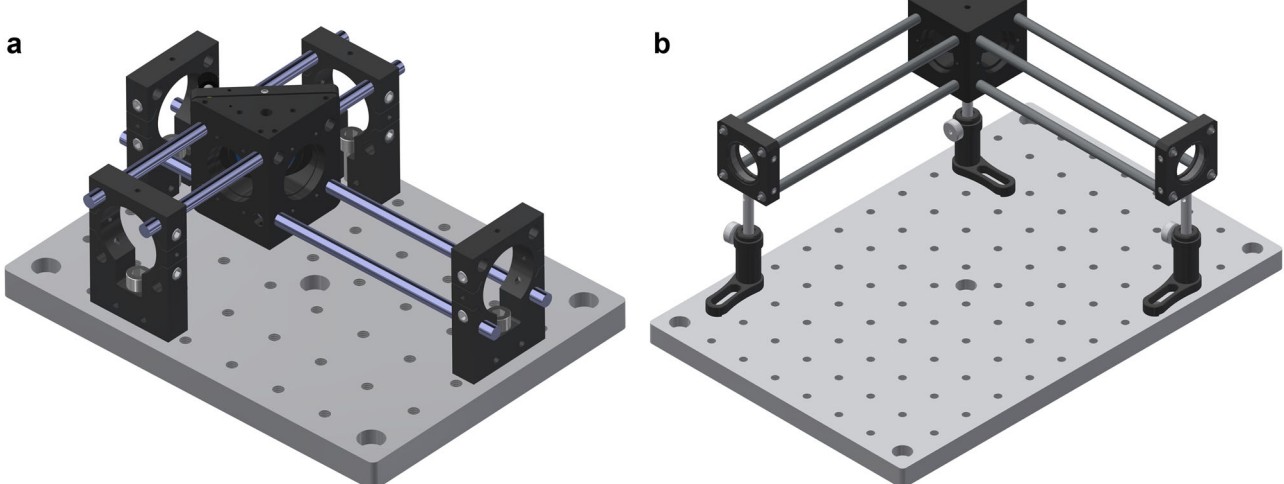

**Fig. 1 | Three-dimensional (3D) computer-aided design (CAD) model of the right-angle kinematic mirror mount assembly.** The assembly is constructed using either **a** ROCS or **b** conventional optical cage systems.

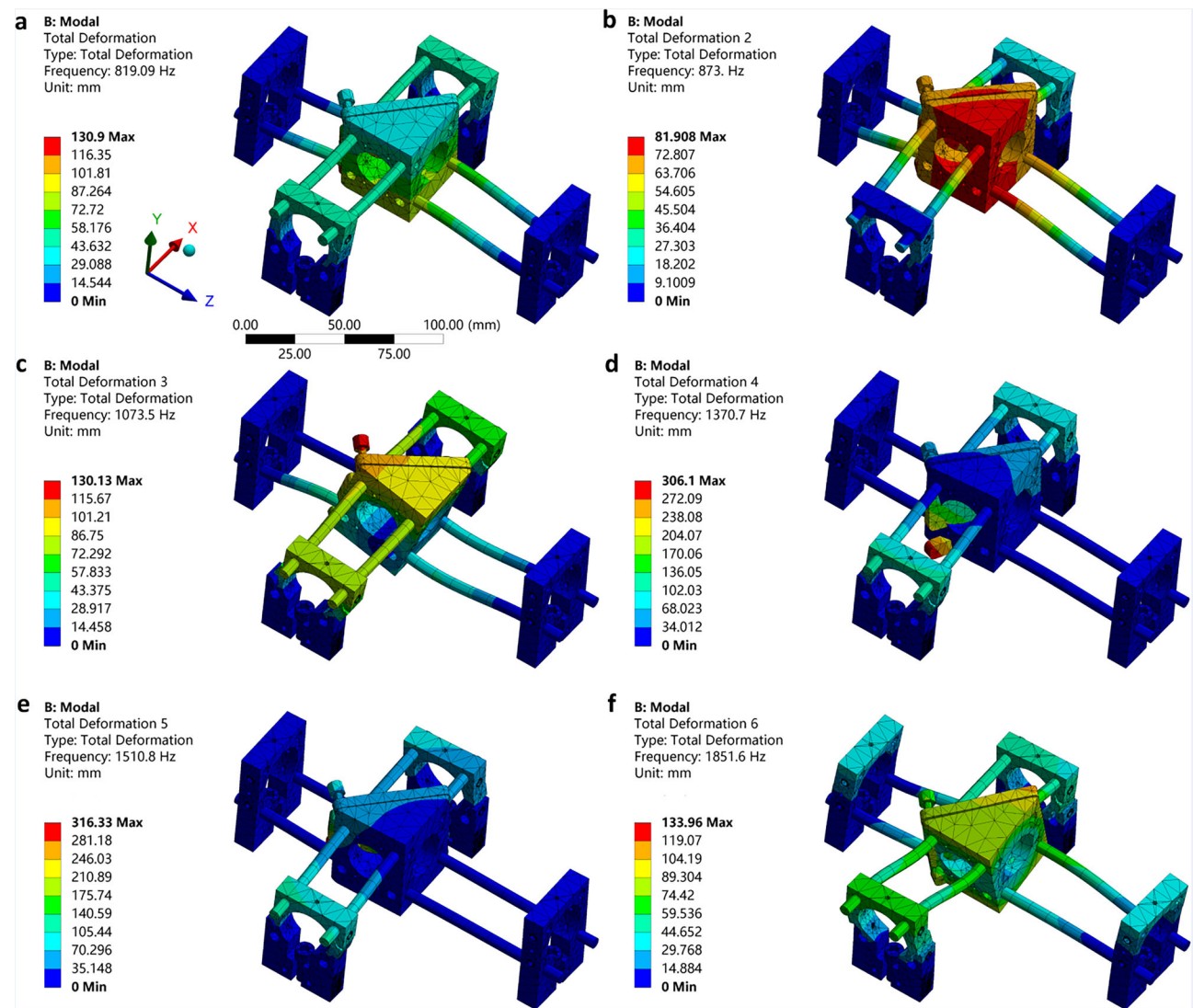

**Fig. 2 | Vibration modal analysis of the right-angle kinematic mirror mount assembly using the ROCS. a–f** The first six modes of the natural frequencies and corresponding mode shapes of the assembly.

By analyzing mode shapes, we can observe how different parts of the structure deform during vibration, offering insights into localized stiffness variations. From the modal analysis results (Fig. 2), first, the presence of distinct and well-separated natural frequencies indicated that the system was capable of effectively resisting various types of vibrational excitation, as the separation minimized the likelihood of resonance under vibrational excitation[28]. Second, the high natural frequency values suggested that the stiffness of the system was sufficient to prevent resonant vibrations within the expected operational range, as systems with higher stiffness exhibit higher natural frequencies[29]. Finally, the absence of modes with frequencies close to 0 implied that there are no substantial rigid-body modes, further reinforcing the system's mechanical stability[27]. This comprehensive analysis confirmed that the system demonstrated excellent mechanical stability, ensuring reliable performance under a range of operational conditions.

In contrast, vibration modal analysis of the right-angle kinematic mirror mount assembly using conventional cage systems revealed several key differences (Fig. S1). Most importantly, the natural frequencies were not well separated. For example, the difference between the first and second modes was 4.47 Hz, compared to 53.91 Hz in the ROCS, revealing a 12-fold difference. Furthermore, the natural frequency values were lower than those in the ROCS overall. For instance, the first mode natural frequency was 317.41 Hz, compared to 819.09 Hz in the ROCS, representing a 2.6-fold

difference. These results indicated significantly lower mechanical stability in the conventional optical cage system.

## Experimental setup

A bench-top microscope was designed and constructed based on ROCS (Fig. 3, Figs. S2, S3 and Supplementary Movie 1). The microscope employed a standard inverted epi-fluorescence microscopy system architecture, and all lenses were arranged as 4 $f$ systems[30] (Fig. 3a). In the illumination optical path, a multi-mode fiber was deployed to convert the Gaussian beam profile of the lasers to a top-hat beam profile, facilitating uniform illumination across the entire field of view. The reflective mirrors, used for beam steering, were housed in right-angle kinematic mirror mounts that were assembled as depicted in Fig. 1a. All lenses were mounted in φ30 mm mounting plates that could be translated along the rods. Two 1.5-inch-diameter optical posts were vertically mounted on the optical bench, serving as a reference for the vertical optical path. The sample stage sub-system (Fig. 3c), which is crucial for the high mechanical stability of the microscope, consisted of two groups of four tungsten-steel rods that formed the backbones of the sub-system. These rods supported and interconnected the optomechanical components in both horizontal and vertical directions. Horizontally, the rods were secured at both ends by two φ30 mm cage plates mounted on the optical bench. Vertically, the rods were held by three φ60 mm cage plates mounted

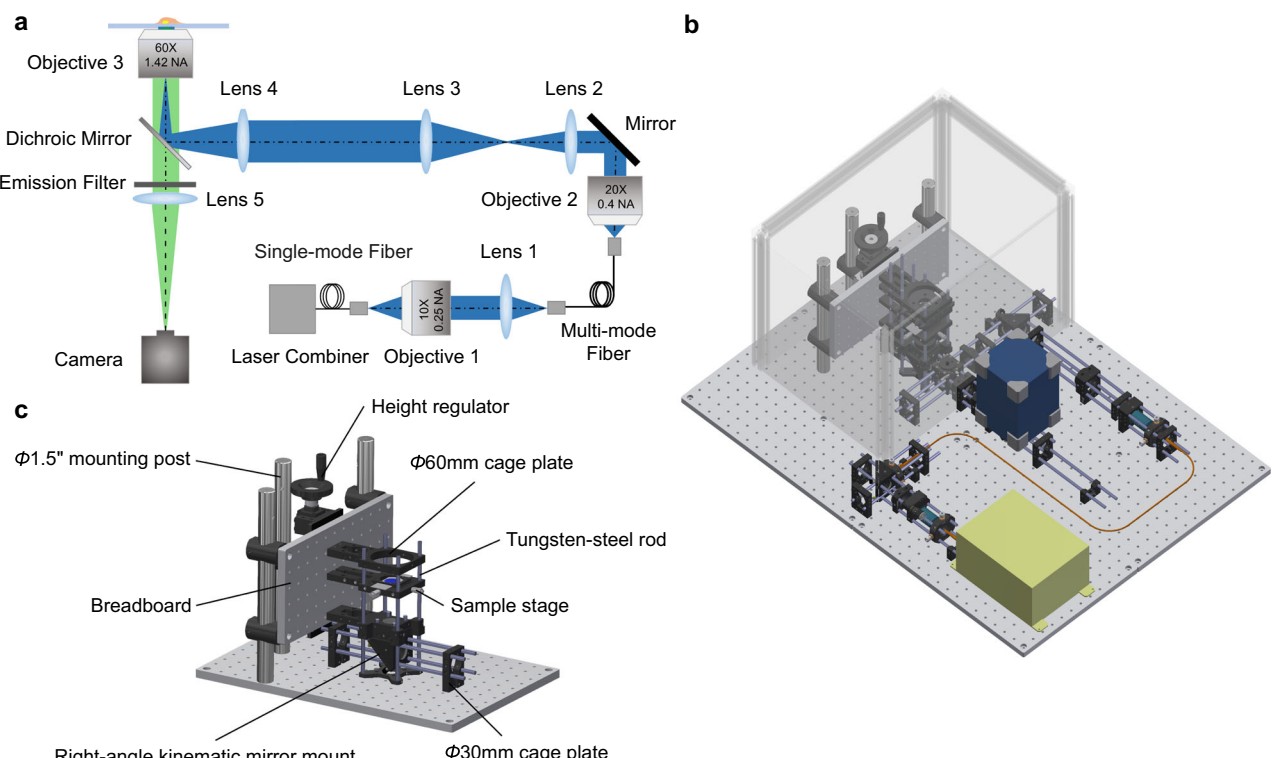

**Fig. 3 | ROCS microscope. a** Schematic of the optical setup. Three-dimensional (3D) computer-aided design (CAD) rendering of **b** the microscope and **c** the sample stage sub-system.

on a vertical breadboard. One of these φ60 mm cage plates also served as the sample holder, incorporating a translation stage equipped with two micrometer screws for positioning samples in the horizontal plane. Coarse axial positioning of samples was achieved using a height regulator that moved the vertical breadboard. Fine axial focusing was accomplished with a piezoelectric stage mounted on the objective lens beneath the cage plate (Figs. S2, S3; see more detail in Methods: Optical setup).

**Evaluation of sample drift in conventional widefield fluorescence microscopy**

To investigate the mechanical stability of the ROCS microscope, we carried out fluorescence microscopy of 100 nm diameter fluorescent beads. The centroid position and PSF size of each fluorescent bead in the 240-frame image sequence, which was captured in 2 h, was determined using the Gaussian fitting function for localization in ThunderSTORM[31], an ImageJ plugin[32]. The changes of the mean position of the bead centroids across the image sequence were used to quantify lateral drift over time, while the changes in the full width at half-maximum (FWHM) of PSFs were used to evaluate axial drift. The experiment was repeated five times on different days, each with fresh immersion oil and at a different location on the coverslip, resulting in five independent image sequences.

To evaluate the axial drift, the changes in the FWHM of PSFs over time in a representative image sequence (Fig. 4a) are presented in Fig. 4b where the difference in PSF size for each frame, relative to the first frame, has been plotted. A 95% confidence interval has also been plotted from the distribution of PSF sizes in the first frame to visualize which frames exhibit axial drift beyond measurement uncertainty. From this plot, the mean change in the PSF FWHM was −3.4 ± 18.6 nm while PSF size estimates for all other frames lay within the confidence interval from the first frame, showing that the changes in PSF size can be attributed to measurement error rather than axial drift. Finally, the mean PSF size change over five experimental iterations was 2.0 ± 5.6 nm (Fig. 4c, see more information in Fig. S4). The analysis of the PSF size over the two-hour period shows no obvious changes, indicating there was no measurable axial drift during this time. After this measurement,

we subsequently repeated the measurement with autofocus enabled to eliminate axial drift through active stabilization of the focal plane using a piezoelectric-driven objective lens, providing a benchmark against the standard measurement result in which the autofocus was disabled (See more information in Methods: Imaging fluorescent beads in the ROCS microscope). Using the same analysis approach, the mean change in PSF size was 0.7 ± 2.1 nm over five experimental iterations (Figs. S5, S6). The previous standard measurement result is comparable to this benchmark, further confirming that the ROCS microscope exhibited no measurable axial drift.

As a direct comparison, we evaluated the axial drift of an off-the-shelf microscope by conducting the same experiment under identical imaging conditions (See more information in Methods: Imaging fluorescent beads in the ZEISS AxioObserver Z1 microscope). Without active focal-plane stabilization, the system exhibited visible axial drift within just 5 min. Following the same analysis approach, the mean change in PSF size was 148.2 ± 54.3 nm for five experimental iterations (Fig. S7). By contrast, the ROCS microscope exhibited a PSF size change of 2.0 ± 5.6 nm in 2 h, demonstrating an exceptional level of mechanical stability compared with conventional commercial microscopes.

Next, we quantified the lateral drift in the ROCS microscope using the same data presented in Fig. 4 and S4. In the lateral directions, the drift trajectories were characterized, and the mean drift values derived from the sequences were calculated, providing quantitative metrics for the evaluation of sample drift. The drift trajectory of the beads in a representative image sequence (Fig. 4a) is presented in Fig. 5a where a trajectory curve has been drawn according to the average drifts of a field of beads over time. The distributions of the absolute values of drift along the x-axis and y-axis are also depicted (Fig. 5b), yielding mean drift values of 0.8 ± 0.7 nm and 1.1 ± 0.7 nm along the x-axis and y-axis, respectively. The localization precision histogram demonstrated a mean value of 11.8 ± 4.8 nm (Fig. 5c). The drift measurement results from the five experimental iterations are plotted in Fig. 5d (also see Fig. S8 & Fig. S9), showing an overall mean lateral drift of 2.5 ± 2.0 nm along the x-axis and 4.0 ± 1.6 nm along the y-axis over a two-hour imaging period. These drift values are two orders of magnitude

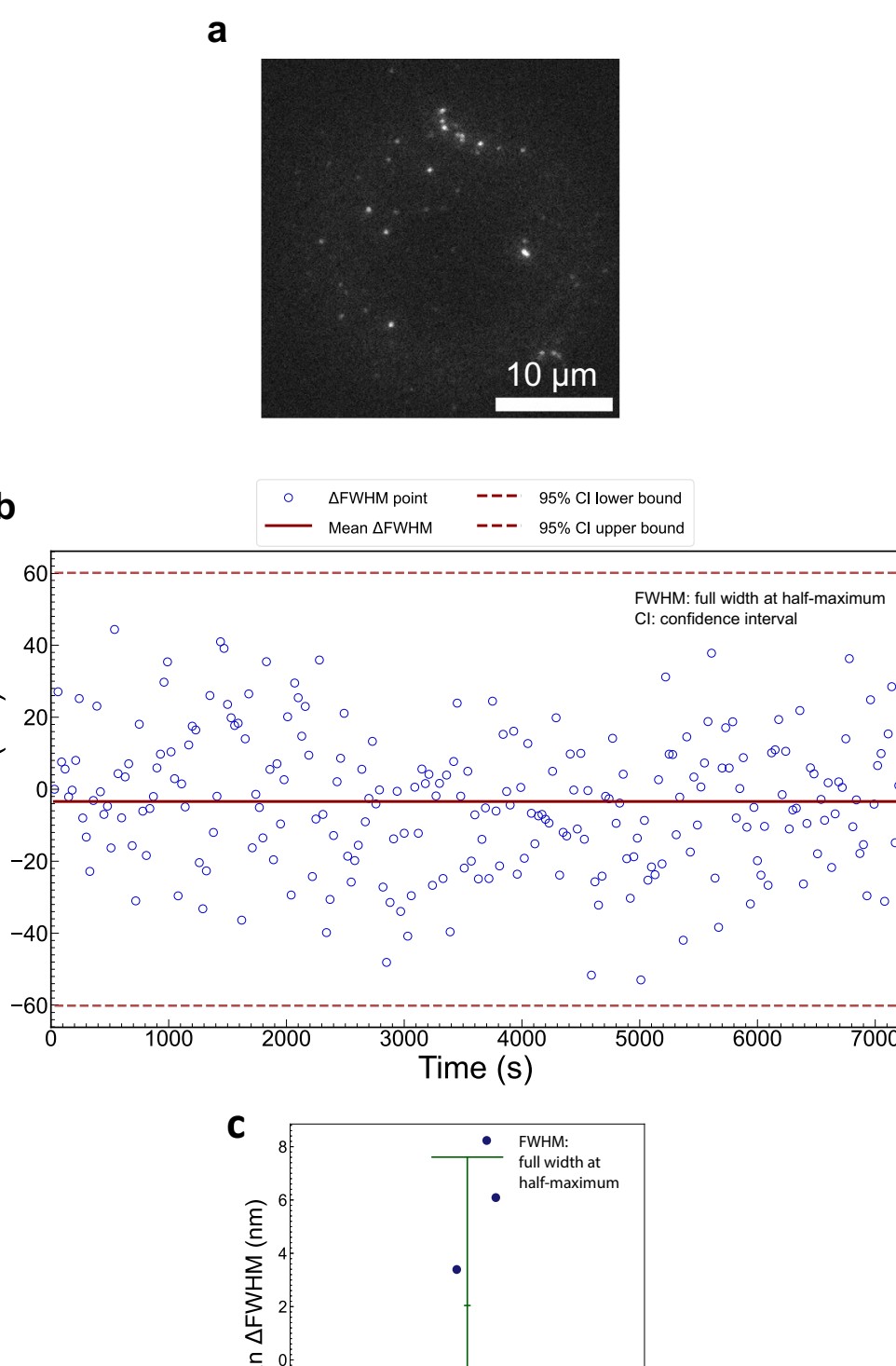

**Fig. 4 | Axial drift evaluation in conventional widefield fluorescence microscopy using the ROCS microscope. a** A representative widefield image of the fluorescent beads. **b** Change in Point Spread Function (PSF) size of the fluorescent beads over 2 h from a representative image series, as measured by the change in the full width at half-maximum (FWHM), i.e., ΔFWHM, of the Gaussian fit. The mean change in the PSF FWHM was −3.4 ± 18.6 nm. The scatter plot of blue circles shows the change in PSF size averaged across all beads in a frame. The solid red line represents the mean change in PSF size across all frames, while the dashed red lines represent the bounds of the 95% confidence interval (CI). **c** Dot plot of mean changes in the PSF FWHM of all five experimental iterations ($n = 5$), showing an overall mean change of 2.0 ± 5.6 nm. The blue dots represent the mean change from each experimental iteration, while the central green bar represents the overall mean change. The upper and lower green bars denote one standard deviation.

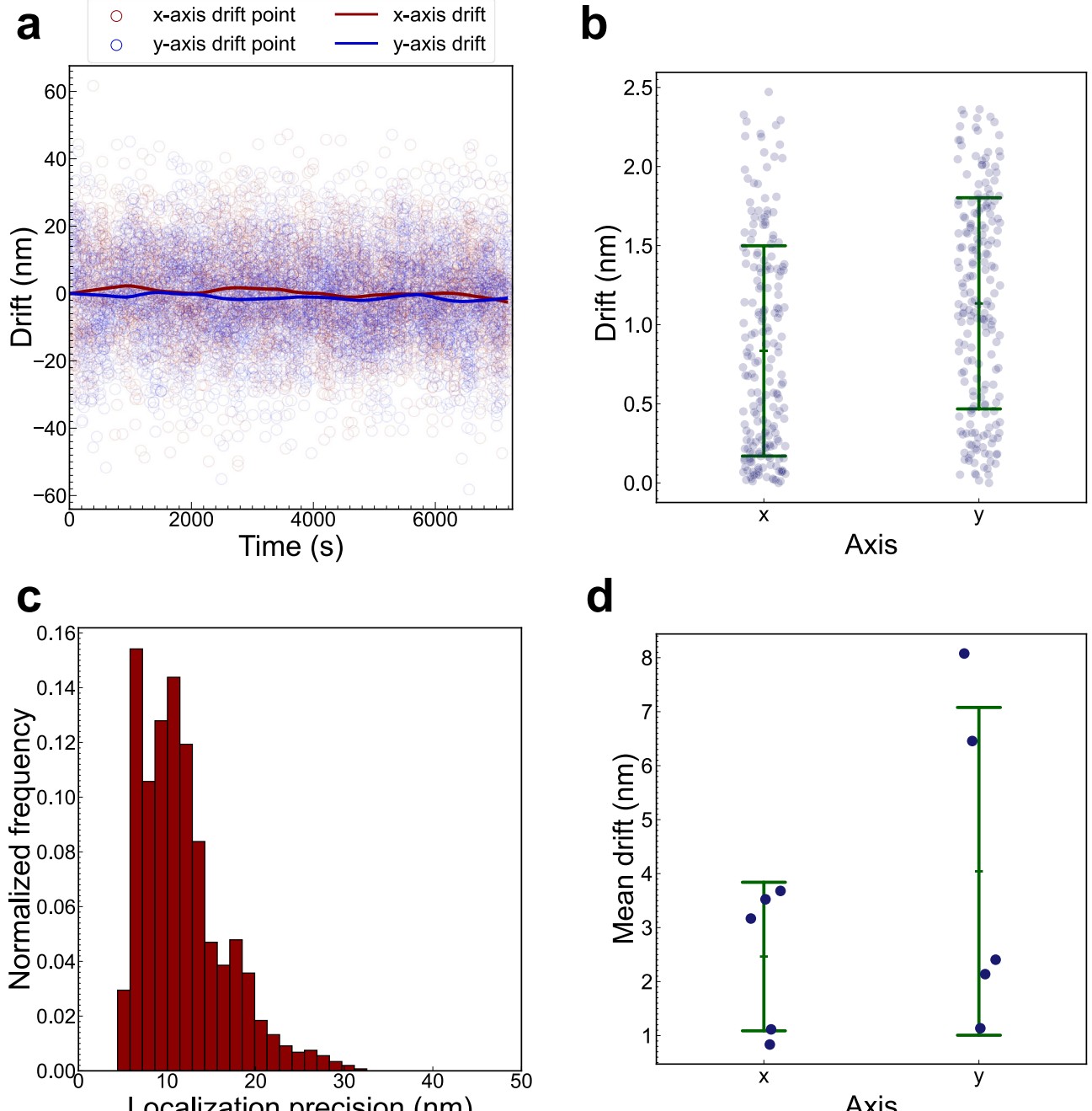

**Fig. 5 | Lateral drift quantification in conventional widefield fluorescence microscopy using the ROCS microscope. a** Drift trajectory of the fluorescent beads over 2 h from a representative image series (Fig. 4a). The scatter plot of red and blue circles shows the positions of the beads which were localized using a Gaussian fit in ThunderSTORM along the x-axis and y-axis, respectively, while the red and blue curves represent the drift trajectories composed of the average drift of all beads as calculated from fiducial marker tracking along the x-axis and y-axis, respectively. **b** Dot plot of the absolute values of the drift along the x-axis and y-axis for the trajectory plotted in (**a**). The blue dots represent the drift trajectory points, the central green bar represents the mean drift, and the upper and lower green bars represent one standard deviation. The mean drift was 0.8 ± 0.7 nm and 1.1 ± 0.7 nm along the x-axis and y-axis, respectively. **c** Localization precision histogram of fluorescent beads in the image series plotted in (**a**). The mean localization precision was 11.8 ± 4.8 nm. **d** Dot plot of the mean drifts from five experimental iterations (n = 5). The mean drift from all iterations was 2.5 ± 1.4 nm along the x-axis and 4.0 ± 3.0 nm along the y-axis. The blue dots represent the mean drifts from each experimental iteration while the central green bar represents the overall mean drift. The upper and lower green bars denote one standard deviation.

smaller than the diffraction-limited resolution, i.e., 200 nm, of widefield fluorescence microscopy. Combined with the unmeasurable drift in the axial direction, these results demonstrate that the ROCS microscope effectively achieved three-dimensional drift-free imaging during extended imaging periods.

Furthermore, we conducted similar measurements with autofocus enabled in the ROCS microscope to investigate the impact of active axial drift correction on lateral drift. Using the same analysis approach, the mean lateral drift was 4.5 ± 2.0 nm along the x-axis and 5.2 ± 1.7 nm along the y-axis over five experimental iterations (Figs. S10–S12). The results are similar to those from the previous standard measurements, demonstrating that active axial drift correction has no measurable impact on lateral drift in the ROCS microscope. More importantly, these findings demonstrate that the ROCS microscope effectively eliminates three-dimensional drift without

the need for active drift stabilization during image acquisition, thereby simplifying hardware requirements and reducing the cost of optical microscopes capable of long-term imaging.

In line with the comparative studies conducted for axial drift characterization, we quantified the lateral drift of an off-the-shelf microscope equipped with active focal-plane stabilization by replicating the same experiment under identical imaging conditions. Using the same analysis method, the mean lateral drift was 148.3 ± 119.8 nm along the x-axis and 259.4 ± 124.1 nm along the y-axis across five experimental iterations (Figs. S13–S15). In contrast, the ROCS microscope exhibited the lateral drift values of 2.5 ± 2.0 nm along the x-axis and 4.0 ± 1.6 nm along the y-axis without active focal-plane stabilization, demonstrating an exceptionally higher level of mechanical stability compared with conventional commercial microscopes.

In the comparative analysis of the ROCS microscope against a conventional microscope, the ROCS microscope exhibited a level of mechanical stability that significantly surpassed that of its conventional counterparts across five experimental iterations. More specifically, the ROCS microscope demonstrated no measurable axial drift and achieved an approximately 60-fold reduction in lateral drift. The experimental evidence gathered from this investigation confirmed that the ROCS microscope achieved nanometer-scale sample drift over extended durations within a conventional wide-field microscopy framework, thereby establishing a foundation for achieving drift-free super-resolution localization microscopy.

## Evaluation of sample drift in STORM

Prior characterization of the ROCS microscope confirmed no detectable axial drift over two-hour data acquisition in conventional wide-field fluorescence microscopy. Since STORM imaging requires a shorter time window, e.g., 15 min, for data acquisition, we infer that axial drift during STORM imaging would remain negligible. Crucially, the system's depth of field (DOF) of 500 nm defines the axial threshold beyond which axial drift becomes resolvable. To degrade STORM localization precision, the focal plane would need to shift by nearly the full DOF within the acquisition window, which is a scenario inconsistent with our long-term stability assays. Furthermore, thermal equilibration of the oil-immersion objective and stage components was maintained across modalities, minimizing differential expansion effects. Therefore, lateral drift evaluation becomes the primary focus of our sample drift analysis in STORM imaging on the ROCS platform.

We implemented three independent repeats of STORM imaging of the microtubule network in fixed HeLa cells using the ROCS-STORM system. Microtubules are polymers of tubulin that associate to form hollow fibrous filaments with a diameter of approximately 25 nm[33], and have been widely used as biological reference standards for quantifying the resolution in STORM[34]. We immunolabeled tubulin with an anti-α-tubulin primary antibody and a secondary antibody conjugated to Alexa Fluor 647 (AF647) dyes.

To investigate the impact of any potential sample drift on the resolution, we applied drift correction on the STORM image using redundant cross-correlation (RCC) in ThunderSTORM. We firstly compared the STORM image quality before and after drift correction qualitatively. Then, we characterized the drift trajectory and calculated the mean drift along the x-axis and y-axis. Finally, we compared the STORM image resolution before and after drift correction using the Fourier ring correlation (FRC) method with a threshold of 0.143[35].

In the first STORM image acquired from the ROCS microscope, details of the microtubule network indiscernible in a conventional widefield image (Fig. 6a) could be clearly discerned in the STORM images with (Fig. 6c) or without (Fig. 6b) drift correction, which is supported by the localization precision of 17.5 ± 7.3 nm in the STORM image (Fig. 6h). It is noteworthy that consistently high localization precision obtained across the image series demonstrates that axial drift was not present in the STORM imaging. The mean drift over 15 min (33000 frames) was found to be 17.0 ± 7.6 nm along the x-axis and 11.9 ± 9.9 nm along the y-axis (Fig. 6f, i). Since the sample

drift was smaller than the mean localization precision, it was reasonable to expect that sample drift contributed insignificantly to the resolution in STORM imaging. As anticipated, the FRC resolution was recorded at 90.4 nm and 83.0 nm prior to and following the implementation of drift correction, respectively (Fig. 6g). In the other two independent STORM imaging experiments, similar values were obtained for the evaluation metrics (Figs. S16, S17). Across all three iterations, the mean drift was 14.4 ± 8.2 nm along the x-axis and 7.3 ± 6.5 nm along the y-axis (Fig. 6i). The resolution remained practically unchanged before and after drift correction in all three iterations, suggesting that the ROCS-STORM system exhibited negligible sample drift during image acquisition. Notably, the sample drift was less than half of the resolution, hence drift correction proved unnecessary in ROCS-STORM imaging[20].

As a comparison, we carried out three independent repeats of STORM imaging on the same sample using an off-the-shelf STORM system. In the first STORM imaging experiment, details of the microtubule network indiscernible in the widefield image (Fig. S18a) were not clearly discerned in the simply reconstructed STORM image (Fig. S18b) but became discernable after cross-correlation drift correction (Fig. S18c). The mean localization precision was 19.0 ± 5.2 nm (Fig. S18h) which was similar to ROCS-STORM; however, the mean drift was markedly larger (Fig. S18f, i), with values of 37.5 ± 19.4 nm along the x-axis and 125.8 ± 67.1 nm along the y-axis. The effect of the larger drift on the resolution was evidenced by the observation that the FRC resolution (Fig. S18g) was improved approximately 6-fold from 312.5 nm to 50.9 nm after implementing cross-correlation drift correction. In the other two independent STORM imaging experiments, similar values were obtained for the evaluation metrics (Figs. S19, S20). Across all three iterations, the mean drift was 143.6 ± 124.0 nm along the x-axis and 93.7 ± 60.0 nm along the y-axis (Fig. S18i). These results contrast those from ROCS-STORM, where the FRC resolution was similar following drift correction, highlighting the occurrence of considerable sample drift in a standard microscope and its detrimental effect on STORM image resolution. The STORM data and drift quantification corroborate the hypothesis that our novel ROCS-STORM confers minimal sample drift, which is unachievable in standard STORM systems.

Apart from microtubules, we also implemented STORM imaging of fixed U2OS cells that endogenously expressed Nup96, a nuclear pore complex (NPC) protein, fused to a SNAP-tag[36] and labeled with Alexa Fluor 647. NPCs are also a common biological reference standard for benchmarking STORM because they have a well-defined, ring-shaped symmetrical structure that serves as a convenient visual indicator of image resolution[37]. We employed the same experimental methodology as that used for microtubule imaging and conducted STORM imaging of NPCs with the ROCS microscope and an off-the-shelf microscope.

In the STORM image acquired from the ROCS microscope (Fig. 7b), the mean localization precision was 13.6 ± 6.6 nm (Fig. 7h) and there was minimal alteration in image quality following drift correction (Fig. 7c), thereby indicating that any residual sample drift did not adversely affect STORM. It is shown that the mean drift was 9.9 ± 6.1 nm and 10.3 ± 5.4 nm along the x-axis and y-axis, respectively (Fig. 7f, i). The comparable FRC resolutions of 66.4 nm without drift correction and 64.1 nm with drift correction (Fig. 7g) indicate that sample drift had a minimal impact. In the other two independent STORM imaging experiments, similar values were obtained for the evaluation metrics (Figs. S21, S22). Across all three iterations, the mean drift was 6.8 ± 5.2 nm along the x-axis and 8.2 ± 5.2 nm along the y-axis (Fig. S18i). These measurements further illustrate the exceptional mechanical stability provided by the ROCS.

We carried out STORM imaging of the same sample on the same off-the-shelf microscope. Again, the ring-shaped structures of NPCs can only be identified following the drift correction (Fig. S23d, e). Compared to the STORM imaging using the ROCS microscope, the mean localization precision was 17.7 ± 6.6 nm (Fig. S23h), which was similar, while the mean drift was higher, with values of 53.4 ± 29.8 nm and 135.2 ± 65.7 nm along the x-axis and y-axis, respectively (Fig. S23f, i). Drift correction led to an approximately fivefold improvement in the FRC resolution, i.e. from

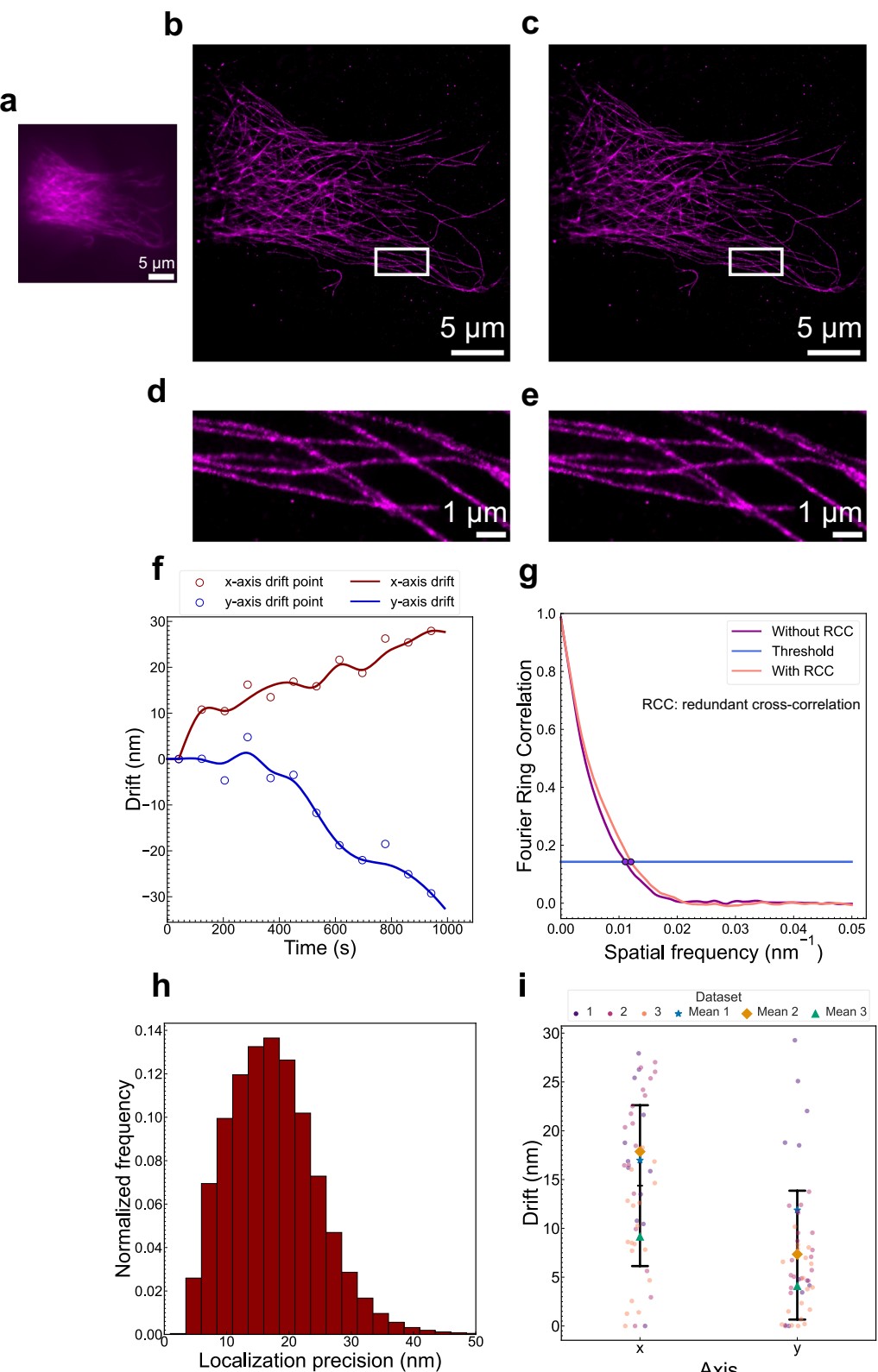

330.2 nm to 64.6 nm (Fig. S23g), representing a significantly larger improvement than that observed with the ROCS-STORM system. In the other two independent STORM imaging experiments, similar values were obtained for the evaluation metrics (Figs. S24, S25). Across all three iterations, the mean drift was 89.3 ± 62.0 nm along the x-axis and 82.0 ± 59.6 nm along the y-axis (Fig. S23i).

The STORM imaging and data analysis of microtubules and NPCs confirmed that conventional microscopes introduce substantial sample drift, degrading image resolution and fidelity. In contrast, and consistent with our FEA modeling and analysis predictions, the ROCS microscope minimized sample drift in STORM, enabling drift-correction-free super-resolution microscopy.

**Fig. 6 | STORM imaging of α-Tubulin-AF647 in the microtubule network in fixed HeLa cells using the ROCS-STORM system. a** Widefield image. **b** STORM image without drift correction and **c** drift-corrected STORM image using redundant cross-correlation (RCC). **d** and **e** Magnified view of the boxed area in **b** and **c**. **f** Drift trajectory of the image series from STORM over about 15 min (33000 frames). The scatter plot of red and blue circles shows the drift data points along the x-axis and y-axis, respectively, while the red and blue curves represent the drift trajectories along the x-axis and y-axis, respectively. The trajectory was calculated using RCC with each circle representing the drift of one bin, i.e. the drift of one image subset relative to the first subset. A total of 12 bins were used. **g** Fourier ring correlation (FRC) resolution evaluation of the STORM images with (**c**) and without (**b**) drift correction, demonstrating 90.4 nm and 83.0 nm resolution before and after drift correction, respectively. A threshold of 0.143 was used. **h** Localization precision histogram of the image shown in **b**. The mean localization precision was $17.5 \pm 7.3$ nm. **i** Dot plot of the mean drifts from three experimental iterations of biologically independent samples (n = 3). The dots represent the absolute values of drift data points and are colored according to their dataset. The central black bar represents the mean drift of all three iterations while the upper and lower black bars represent one standard deviation. The mean drift of each dataset is represented by a different shape. The mean drift of the first dataset shown above was $17.0 \pm 7.6$ nm along the x-axis and $11.9 \pm 9.9$ nm along the y-axis. The mean drift from all iterations was $14.4 \pm 8.2$ nm along the x-axis and $7.3 \pm 6.5$ nm along the y-axis.

## Discussion

Conventional optical cage systems are widely used in optical instrument prototyping due to their compact design, precise alignment of optical components, straightforward optomechanical integration, and simple modular scalability. The ROCS not only retains these advantages but also offers unprecedent mechanical stability. In bench-top microscopy systems utilizing conventional optical cage designs, sample drift primarily arises from system instability caused by localized shear, tensile, and compressive stresses at multiple discrete optomechanical components. Our vibration modal analysis, conducted through FEA, reveals that the use of ROCS effectively eliminates rigid-body displacement modes, thereby significantly enhancing system stability. In contrast, commercially available stand-alone microscopes typically employ a robust iron-cast chassis to provide a high degree of structural stability. However, as demonstrated in our experiments, these systems still experience considerable sample drift due to instability resulting from the translation stage assembly mounted on top of the microscope chassis. Through both simulations and experimental validation, we show that microscopy systems employing ROCS demonstrate superior system stability compared to those utilizing conventional optical cage systems or commercial microscopes. A notable application is the high-performance ROCS-STORM system, which eliminates the need for sample drift correction. Using the ROCS-STORM system, we achieved resolutions of a few tens of nanometers in biological imaging, without requiring additional hardware, fiducial markers in the samples, or image post-processing. As a comparison, some bench-top STORM systems constructed with conventional optical cage systems[38,39] exhibited sample drift that was comparable to the drift measured in the off-the-shelf microscope of this study, demonstrating the limited mechanical stability of conventional optical cage systems-based assemblies in contrast to the superior stability demonstrated by the ROCS. This work marks the first demonstration of the ROCS's capability for super-resolution microscopy with exceptional mechanical stability. The total cost of the ROCS-STORM system was approximately US$78,000 (see Table S1), which is comparable to that of other low-cost bench-top STORM system constructed with conventional optical cage systems[39], and substantially lower than the cost of commercially available instruments. ROCS offer enhanced ease of assembly over conventional optical cage systems through an integrated reinforced framework and modular, perforated cage components. In contrast to conventional optical cage systems, where optomechanical components are rigidly fixed to stainless steel rods using screws, ROCS allow unrestricted translation of optomechanical components along tungsten steel rods. This flexibility reduces assembly time and facilitates rapid reconfiguration of optical setups. Other advantages of the ROCS have also been shown to improve biological imaging, including applications in Mueller microscopy[40] and digital holographic microscopy[41]. Beyond biological imaging, ROCS is poised to benefit engineering fields where instrument stability becomes critically important.

While the ROCS offers exceptional mechanical stability, it may not be suitable for all scientific instrumentation due to its trade-off in reduced degrees of freedom, similar to conventional optical cage systems. This limitation in reduced flexibility may restrict its application in systems requiring free-space configurations, such as some off-axis optical setups. Developing a more adaptable ROCS design to address these needs is in the development pipeline of our future work.

In this study, microtubules and NPCs in mammalian cells were used as standard biological model systems for benchmarking. To comprehensively evaluate the performance and versatility of the ROCS-STORM system, future research should include testing on biological systems with a broader variety of cellular structures.

Additionally, the FRC resolution of the STORM images obtained in this work for biological imaging could be further improved, since higher resolutions of STORM images have been previously reported for both microtubules and nuclear pore complexes labeled with Alexa Fluor 647[34,37]. Achieving higher resolution requires improvements in factors, such as localization precision, labeling density, and preserving the structure of the sample[35]. Although optimization of resolution was beyond the scope of this study, the biological imaging data acquired using the ROCS-STORM system allowed us to analyze and quantify sample drift, demonstrating that the small amount of residual drift had negligible effects on resolution.

## Conclusions

We introduce a novel approach based on ROCS to achieve mechanically drift-free SMLM, offering a straightforward and effective alternative to existing drift-correction methods. By negating sample drift caused by mechanical instability, our method eliminates the need for additional hardware, sample preparation, or image post-processing, thereby simplifying experimental setups and workflows, and reducing the cost. In conventional wide-field microscopy, the ROCS microscope exhibited excellent stability in three dimensions, with the lateral drift of samples limited to $2.5 \pm 1.4$ nm along the x-axis and $4.0 \pm 3.0$ nm along the y-axis, and without any measurable axial drift over 2 h. In STORM, the ROCS microscope maintained minimal sample drift, which was $14.4 \pm 8.2$ nm along the x-axis and $7.3 \pm 6.5$ nm along the y-axis over 15 min of imaging microtubules, and $6.8 \pm 5.2$ nm along the x-axis and $8.2 \pm 5.2$ nm along the y-axis for nuclear pore complexes, without compromising resolution performance. The lateral drift in STORM was greater than that observed in conventional widefield microscopy due to the thermal expansion resulting from the higher laser power density ($>1$ kW/cm$^2$) used during STORM imaging (Fig. S26). The stability of the system within the ROCS microscope has constrained sample drift to a value significantly less than half the resolution, thus making drift correction unnecessary in STORM. Consequently, we have successfully attained a state of drift-free STORM. This methodology serves as a paradigm for the application of ROCS in SMLM techniques, providing exceptional drift-free imaging capability in the field of super-resolution microscopy. In the broader context of engineering, the ROCS is positioned to become a cornerstone for prototyping highly reliable and ultra-stable scientific instruments, transforming the standards of precision and stability in advanced instrumentation.

## Methods
### Finite Element Analysis

In designing the ROCS microscope, we used Autodesk Inventor software (Autodesk, San Rafael, USA) to simulate the microscope assembly, and then imported the model into ANSYS FEA software (ANSYS, Canonsburg, USA) to conduct vibration modal analysis. In the simulation, a finite element mesh model was built under the following conditions:

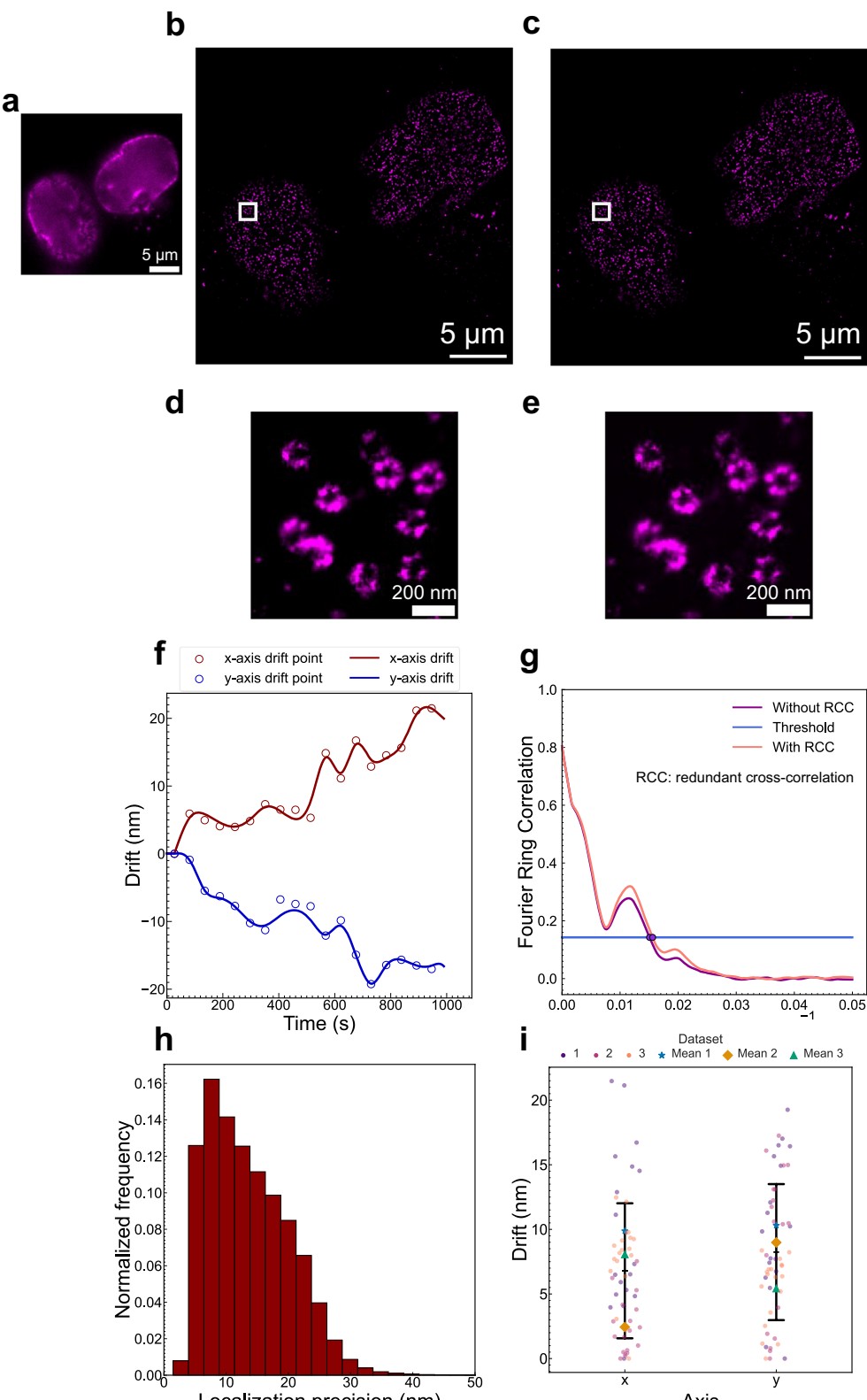

1. During the preprocessing stage, components that contribute minimally to structural integrity, such as small screws and springs, were removed. Additionally, any damaged surfaces were repaired as needed. This step aimed to simulate the complete structure in real-world conditions, minimizing potential errors and improving the accuracy of subsequent model calculations.

2. The material properties and boundary conditions were defined as follows. All mechanical components used in the ROCS, except for the cage assembly rods, were made of aluminum alloy 6061-T6, which has an elastic modulus of 68.9 GPa and a Poisson's ratio of 0.33[42]. The cage assembly rods were made of tungsten steel, which has an elastic modulus of 410 GPa and a Poisson's ratio of 0.28[43].

**Fig. 7 | STORM imaging of Nup96-SNAP-AF647 in fixed U2OS cells using the ROCS-STORM system. a** Widefield image of two nuclei. **b** STORM image without drift correction and **c** drift-corrected STORM image using redundant cross-correlation (RCC). **d** and **e** Magnified view of the boxed area in **b** and **c**. **f** Drift trajectory of the image series over about 15 min (33000 frames). The scatter plot of red and blue circles shows the drift data points along the x-axis and y-axis, respectively, while the red and blue curves represent the drift trajectories along the x-axis and y-axis, respectively. The trajectory was calculated using RCC with each circle representing the drift of one bin, i.e. the drift of one image subset relative to the first subset. A total of 18 bins were used. **g** Fourier ring correlation (FRC) resolution evaluation of the STORM images with (**c**) and without (**b**) drift correction, demonstrating 66.4 nm and 64.1 nm resolution before and after drift correction, respectively. A threshold of 0.143 was used. **h** Localization precision histogram of the image shown in **b**. The mean localization precision was 13.6 ± 6.6 nm. **i** Dot plot of the mean drifts from three experimental iterations of biologically independent samples (n = 3). The dots represent the absolute values of drift data points and are colored according to their dataset. The central black bar represents the mean drift of all three iterations while the upper and lower black bars represent one standard deviation. The mean drift of each dataset is represented by a different shape. The mean drift of the first dataset shown above was 9.9 ± 6.1 nm along the x-axis and 10.3 ± 5.4 nm along the y-axis. The mean drift from all iterations was 6.8 ± 5.2 nm along the x-axis and 8.2 ± 5.2 nm along the y-axis.

3. In the global structural analysis, the constraints imposed were consistent with real-world conditions. For instance, constraints were applied where there were threaded connections and tungsten-steel rods passed through the optomechanical components.

4. A standard gravitational field was applied in the simulation.

5. Nodes in the finite element mesh were generated to discretize the geometry of the object, forming a detailed mesh for analysis. Using default settings for meshing, stress, strain, and displacement were calculated throughout FEA. The stress of each node was simulated, producing a stress distribution contour map. Additionally, the first six vibration modes of the model were extracted, and their corresponding mode shapes were displayed as contour plots.

### Fluorescent beads sample preparation
100 nm TetraSpeck fluorescent beads (T7279, Thermo Fisher) were mounted on poly-L-lysine (PLL, Catalog no. P4707, Sigma-Aldrich)-coated coverslips. The coverslips (22 mm × 22 mm, #1.5, Catalog no. 631-0124, VWR) were first covered with 200 μL of 0.01% (w/v) PLL for an hour at ambient temperature. The fluorescent beads were diluted in Milli-Q® water at a 1:100 ratio. Following this, the PLL on the coverslip was removed, replaced with the diluted bead solution, and left at ambient temperature for half an hour. Excess diluted bead solution was removed. 0.5 μL of Milli-Q® water was then added onto a cover slide (76 mm × 26 mm x 1 mm, Catalog no. MIC3028, Scientific Laboratory Supplies, Nottingham, UK), and the coverslip was mounted on the cover slide, with the beads sandwiched between the coverslip and cover slide. Duplicating silicone dental glue (Twinsil 20, Picodent, Wipperfürth, Germany) was applied along the edges of the coverslips to seal the samples. Finally, samples were stored in the refrigerator at 4 °C.

### Cell culture
Cell samples were prepared according to previously published protocols[37,44]. HeLa cells (ATCC CCL-2) were cultured at 37 °C and 5% CO$_2$ in high glucose Dulbecco's Modified Eagle Medium (DMEM, Catalog no. 11965092, Gibco) supplemented with 10% fetal bovine serum. U2OS cells endogenously expressing Nup96-SNAP (Catalog No. 300444, Cell Line Service) were cultured at 37 °C and 5% CO$_2$ in McCoy's 5 A Medium supplemented with 10% fetal bovine serum and 1x non-essential amino acids (Catalog No. 11140050, Thermo Fisher). Prior to seeding, 22 mm × 22 mm high-precision glass coverslips (Catalog No. 631-0124, VWR) were sterilized in 70% ethanol and coated with 0.1% (w/v) poly-L-lysine for 30 min at ambient temperature. Cells were plated on the coverslips and left to grow for 24—48 h before fixation and labeling.

### Immunolabeling of fixed HeLa cells
The seeded cells were pre-fixed for 15—45 s at 37 °C with 0.25% (v/v) Triton and 0.1% (v/v) glutaraldehyde (Catalog No. G5882, Sigma-Aldrich) in PEM (80 mM PIPES, 5 mM EGTA, 2 mM MgCl$_2$, pH 6.8) to remove cytoplasmic proteins. Cells were fixed at 37 °C with 0.25% Triton and 0.5% glutaraldehyde in PEM for 10 min. Subsequently, cells were washed twice in PBS and quenched for 7 min with 0.1% (w/v) NaBH$_4$, followed by blocking and permeabilization in blocking buffer (0.22% [w/v] gelatin, 0.1% Triton X-100, pH 7.3) for at least 1 h at ambient temperature with gentle agitation. The cells were then incubated overnight at 4 °C with anti-α-tubulin primary antibodies (Catalog No. T5168, monoclonal mouse antibody, Sigma-Aldrich) diluted 1:300 in blocking buffer. Following three washes with blocking buffer, cells were stained for 1 h with donkey anti-mouse IgG (H + L) highly cross-adsorbed secondary antibody, Alexa Fluor plus 647 (Catalog No. A32787, Thermo Fisher Scientific) dissolved in blocking buffer at a concentration of 5 μg/mL at ambient temperature. Finally, samples were washed once with blocking buffer for 10 min, then washed twice with PBS to remove unbound antibodies.

### SNAP-tag labeling of fixed U2OS cells
Cells were pre-fixed with 2.4% (w/v) paraformaldehyde (Catalog No. 15710, Electron Microscopy Sciences) for 30 s at ambient temperature before permeabilization for 3 min using 0.4% Triton-X100. Cells were then washed once with PBS and fixed with 2.4% paraformaldehyde for 30 min. The fixation solution was subsequently quenched using 50 mM NH$_4$Cl. Following this, samples were incubated with Image-iT FX Signal Enhancer (Catalog No. I36933, Thermo Fisher) for 30 min and then stained with 1 μM SNAP-Surface Alexa Fluor 647 (Catalog No. S9136S, New England Biolabs) dissolved in PBS with 0.5% (w/v) BSA and 1 mM DTT.

### Sample preparation for STORM
Prior to imaging, concave coverslides (Catalog No. MS15C1, Thorlabs) were washed with 100% ethanol and STORM buffer (50 mM or 100 mM cysteamine hydrochloride [MEA], 0.04 mg/mL catalase, 0.5 mg/mL glucose oxidase, 10% [w/v] glucose dissolved in 50 mM Tris-HCl, 10 mM NaCl, pH 7.5) was prepared. 100 μL of buffer was dispensed onto the coverslide and the coverslips were mounted with the cells sandwiched between the coverslide and the coverslip. Finally, the coverslips were sealed using duplicating silicone dental glue (Twinsil 20, Picodent, Wipperfürth, Germany). STORM imaging was performed in freshly made STORM buffer each time. All chemicals were purchased from Sigma-Aldrich unless otherwise specified.

### Optical setup
In the bench-top ROCS microscope, a laser combiner (iChrome-CLE50, Toptica, Munich, Germany) equipped with 405 nm, 488 nm, 561 nm, and 640 nm laser lines was used as the light source. The laser output was provided via a polarization-maintaining single-mode fiber. The lasers were coupled into an achromatic objective (20x, 0.25NA, Leica, Wetzlar, Germany) and collimated. The collimated beams were focused by an achromatic doublet lens (f = 60 mm) and coupled into a multi-mode fiber (Catalog No. M72L01, Ø200 μm, 0.39NA, FC/PC to FC/PC, Thorlabs, Newton, NJ, USA). The multi-mode fiber was used to transform the Gaussian beam profile from the single-mode fiber into a top-hat beam profile. After that, the lasers were coupled into another achromatic objective (20x, 0.4NA, Olympus, Tokyo, Japan). A pair of achromatic doublet lenses (f = 30 and 50 mm) were used to expand the lasers. The lasers were reflected to a high NA oil immersion objective (60x, 1.42NA, APON 60XOTIRF, Olympus, Tokyo, Japan) using a dichroic mirror (Di01-R405/488/561/635-25 × 36, Semrock, West Henrietta, NY, USA). The collimated lasers were focused onto the back focal plane of the objective after passing through an achromatic lens (f = 250 mm) to produce Köhler illumination onto the sample plane. The sample slides were mounted, using a magnetic holder, onto a custom-designed manual sample stage with a 10 μm translation

range along the x- and y-axes. During image acquisition, the focal plane of the microscope could be optionally maintained by a piezoelectric stage (P-721 PIFOC, Physik Instrumente, Karlsruhe, Germany) mounted underneath the objective. The axial position of the piezoelectric stage was actively adjusted based on the grayscale values of sample images using Micromanager 2.0[45]. The images from the ROCS microscope were saved as 16-bit TIFF files using Micromanager 2.0.

For imaging fluorescent beads with the 488 nm laser, a 525/50 nm emission filter (FF02-525/50-25, Semrock, West Henrietta, NY, USA) was used. For STORM imaging with a 640 nm laser, a 690/50 m emission filter (ET690/50 m, Chroma, Bellows Falls, VT, USA) was used. An achromatic lens (f = 200 mm) focused the imaging beam onto an sCMOS camera (01-Prime-BSI-R-M-16-C, Teledyne Photometrics, Tucson, AZ, USA). The camera chip had $2048 \times 2048$ pixels, with a pixel size of $6.5\ \mu m \times 6.5\ \mu m$. The images projected onto the camera had a pixel resolution of 97.5 nm per pixel, resulting in a $200\ \mu m \times 200\ \mu m$ field of view in the object plane. A $38\ \mu m \times 38\ \mu m$ effective field of view was realized after beam shaping of the lasers. To prevent potential vibrations generated by the fan, the sCMOS camera was water-cooled with the fan disabled. Furthermore, to mitigate the effects of air currents in the laboratory, a blackout enclosure (custom black hardboard, Thorlabs, Newton, NJ, USA), covering the sample stage subsystem completely, was used to prevent air flow and stabilize temperature. All optics were from Thorlabs unless noted otherwise. All ROCS components were from RayCage (Zhenjiang) Photoelectric Technology Co., Ltd., China unless noted otherwise.

### Imaging fluorescent beads in the ROCS microscope
The beads were illuminated using a 488 nm laser at a power density of $0.01\ kW/cm^2$. The fluorescence emitted by the beads passed through the dichroic mirror and a 525/50 m emission filter before being detected by the sCMOS camera. Images of beads were taken every 30 seconds for 2 h with an image exposure time of 100 ms and a maximum readout speed of 95 frames per second in the camera, resulting in an image sequence containing 240 frames.

While collecting image sequences of beads, image acquisition from the camera was synchronized with illumination from the laser. In other words, the laser was only turned on when the camera was acquiring images, minimizing sample drift caused by thermal expansion. A benchmark PSF measurement unaffected by axial drift was obtained using the built-in autofocus function in Micromanager[45,46] in combination with a piezoelectric stage mounted underneath the objective lens to eliminate axial sample drift, whereas in the standard measurement the autofocus function was disabled. The autofocus function captures a series of images along the z-axis—with each image separated by a user-specified distance—after a user-specified number of frames elapse in the image sequence. From this series of images, the sharpest image was identified, and the axial position of the objective lens was adjusted to maintain focus. We applied the autofocus function to every frame. We also allowed a 20-minute set-down time before each image acquisition to stabilize adhesive forces between the immersion oil and coverslip.

### STORM imaging in the ROCS microscope
Samples were excited with a 640 nm laser, initially set at a power density of $4\ kW/cm^2$ to establish a stable blinking state and then reduced to $1.6\ kW/cm^2$ for image acquisition. The emitted fluorescence was directed through the dichroic mirror and a 690/50 m emission filter before detection by the camera. A total of 33,000 images were captured with an exposure time of 30 ms per frame. The autofocus function was disabled during STORM imaging. Prior to STORM image acquisition, a widefield image of the sample was captured with a laser power density of $4\ W/cm^2$ and a camera exposure time of 100 ms.

### Imaging fluorescent beads in the ZEISS AxioObserver Z1 microscope
Fluorescence microscopy was carried out using a ZEISS AxioObserver Z1 microscope equipped with a high *NA* objective (100x, 1.46*NA*, α Plan-

Apochromat 100x/1.46 Oil DIC M27, ZEISS, Oberkochen, Germany). TetraSpeck fluorescent beads were excited using a 488 nm laser at a power density of $0.01\ kW/cm^2$. The fluorescence was first passed through an MBS-488 dichroic mirror (ZEISS, Oberkochen, Germany) and a BP495-550/LP750 emission filter (ZEISS, Oberkochen, Germany). For both axial and lateral drift characterization, image sequences were acquired with an exposure time of 100 ms per frame and a maximum readout speed of 56 frames per second using an EMCCD camera (iXon DU 897, Andor Technology, Belfast, UK). Image acquisition was performed using the Zen software (ZEN 2.3 black edition, ZEISS, Oberkochen, Germany). In the axial drift experiment, the images were taken continuously for 5 min, during which the samples became noticeably out of focus. In the lateral drift experiment, the images were taken every 30 seconds for 2 h, resulting in an image sequence containing 240 frames. Axial drift was actively corrected in real time using the definite focus functionality in the microscope.

### STORM imaging in the ZEISS Elyra PS.1 microscope
STORM was carried out using a ZEISS Elyra PS.1 microscope equipped with a high *NA* objective (100x, 1.46*NA*, α Plan-Apochromat 100x/1.46 Oil DIC M27, ZEISS, Oberkochen, Germany). Samples were excited using the 642 nm laser. The laser power density was first raised to $14\ kW/cm^2$ to achieve a stable blinking state and then lowered to $2.8\ kW/cm^2$ for data collection. The fluorescence was first passed through an MBS-642 dichroic mirror (ZEISS, Oberkochen, Germany), and then detected by a LP655 emission filter (ZEISS, Oberkochen, Germany). 33,000 images in a $25.6\ \mu m \times 25.6\ \mu m$ field of view were recorded with an exposure time of 30 ms per frame and an EM gain of 250 using an EMCCD camera (iXon DU 897, Andor Technology, Belfast, UK). Image acquisition was performed using the Zen software (ZEN 2.3 black edition, ZEISS, Oberkochen, Germany). The axial drift of the samples was corrected continuously in real time using definite focus functionality in the microscope. A wide-field image was captured with a laser power density of $0.1\ kW/cm^2$ and a camera exposure time of 200 ms before the STORM image sequence was collected.

### STORM image processing
The STORM data acquired from both the ROCS microscope and the ZEISS Elyra PS.1 microscope were processed using the same single-molecule localization image processing pipeline. Sub-pixel localization of single fluorophores in the image sequence was carried out using the ThunderSTORM plugin (version 1.3)[31] in ImageJ (version 1.53t)[32]. The camera settings of the ROCS microscope for ThunderSTORM were as follows: a camera offset of 100, a pixel size of 97.0 nm/pixel, and 0.59 photoelectrons per analog-digital unit. The camera settings of the Elyra microscope were: an offset of 100, a pixel size of 100 nm/pixel, 15.3 photoelectrons per analog-digital unit, and an EM gain of 250. The images were first filtered by a B-spline wavelet filter[47] using default settings, i.e., a third-order B-spline basis function and a scaling factor of 2. The approximate locations of fluorophores were then determined by detecting the local intensity maxima. If a pixel had an intensity greater than a user-specified threshold of $1.5\sigma$—where $\sigma$ is the standard deviation of intensity from the first wavelet level—and an intensity greater than or equal to its 8 nearest-neighbors, it was selected for sub-pixel localization.

During sub-pixel localization, the lateral positions of fluorophores were determined by fitting with a 2D integrated Gaussian PSF model via maximum likelihood estimation[48]. The Gaussian PSF standard deviation was 1.62 pixels for all localizations and the fitting radius used to define the approximate location of a fluorophore was 5 pixels for all localizations. The Normalized Gaussian setting was used for visualization with a super-resolution pixel size of 10 nm/pixel. Image resolution was measured using the Fourier ring correlation method in an ImageJ plugin developed by Nieuwenhuizen et al. (2013)[35]. The image resolution was determined when the corresponding spatial frequency achieved a correlation threshold of 1/7. FRC curves were plotted using custom scripts written in Python 3.10 with the NumPy (version 2.1.1) and Matplotlib (version 3.9.2) packages[49,50].

## Drift quantification

The axial drift was quantified from the localization table from Thunder-STORM. For each bead, a measure of the PSF width is given by the standard deviation of the Gaussian function used to fit the PSF. From this standard deviation, the FWHM was calculated as:

$$FWHM = 2\sigma\sqrt{2\ln 2} \qquad (1)$$

where $\sigma$ is the standard deviation of the Gaussian fit. The FWHMs of all beads in each frame were then averaged and the change in FWHM ($\Delta$FWHM) was calculated using the first frame as a reference. A 95% confidence interval was also calculated as:

$$\left[\overline{\Delta FWHM} - t\,\frac{s}{\sqrt{n}}, \overline{\Delta FWHM} + t\,\frac{s}{\sqrt{n}}\right] \qquad (2)$$

$\overline{\Delta FWHM}$ is the mean change in the FWHM for the first frame and therefore equals zero. $t$ is the critical value of the t-statistic, $s$ is the standard deviation of FWHMs for the first frame, and $n$ is the number of beads in the first frame.

The lateral drift of fluorescent beads was quantified using a built-in algorithm from ThunderSTORM that tracks the positions of beads across an image series[31]. The algorithm first identifies fiducial markers automatically from molecular localizations that fluoresce at one position for a substantial amount of time. The drift trajectory data point for a single frame is then calculated by averaging the positions of all identified fiducial markers according to the following equation:

$$\bar{x}_t = \frac{1}{M}\sum_{i=1}^{M}(x_{i,t} - \theta_i) \qquad (3)$$

here, $\bar{x}_t$ denotes the average drift, $M$ is the number of fiducial markers in a frame, $x_{i,t}$ is the absolute position of the $i$-th marker at frame $t$, $i = 1, \ldots, M$, and $\theta_i$ is an unknown offset.

The offset is estimated through least squares minimization of the sum of squared differences between the relative sample positions and the average sample drift, $\bar{x}_t$, as defined by the equation:

$$\hat{\boldsymbol{\theta}} = \sum_{t=1}^{T}\sum_{i=1}^{M}\left((x_{i,t} - \theta_i) - \bar{x}_t\right)^2 \qquad (4)$$

$\hat{\boldsymbol{\theta}} = [\hat{\theta}_i, \ldots, \hat{\theta}_M]$ is the estimated offset for each fiducial marker in the time series. From these calculations, the mean drift was quantified by taking the absolute value and averaging the drift trajectory data points over time, i.e., by averaging all values of $|x_{i,t}|$. The positions of beads were plotted as a scatter plot while the drift trajectory, derived from the average drift in each frame, $\bar{x}_t$, was plotted as a curve.

Drift from STORM was quantified using RCC in Super-resolution Microscopy Analysis Platform (SMAP) software[22,51]. RCC involves dividing the image sequence into a user-specified number of subsets and comparing the detected sample structure from one subset to all other subsets. Through this comparison, the changes in fluorophore positions over time were tracked, and so an estimate of the drift over time was acquired. The mean drift was calculated by averaging the absolute values of the drift of each image sequence subset along the x-axis and y-axis. All drift trajectories were plotted using custom scripts written in Python 3.10 with the NumPy (version 2.1.1), Matplotlib (version 3.9.2), Pandas (version 2.2.2), and Seaborn (0.13.2) packages[52,53].

## Reporting summary

Further information on research design is available in the Nature Portfolio Reporting Summary linked to this article.

## Data availability

The datasets generated and analyzed in this study are available in the STFC eData repository, [https://doi.org/10.5286/edata/942], with additional datasets in the Zenodo repository, [https://doi.org/10.5281/zenodo.15407010].

## Code availability

The code used in this study is available from the corresponding author upon reasonable request.

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

## Acknowledgements

This work was supported by Rosetrees Trust and The Stoneygate Trust grant (Seedcorn2022\100230), STFC Proof of Concept Fund award (POCF2021-14), STFC Cancer Diagnosis Network+ grant (ST/S005404/1), and Biotechnology and Biological Sciences Research Council (UKRI-BBSRC) grant (BB/T008784/1). STFC Central Laser Facility also funded this work. We thank Christopher J. Tynan for providing some fixed U2OS cells expressing Nup96-SNAP-AF647.

## Author contributions

L.W. conceived the project. H.Q., M.C.T., S.L., X.L., and L.W. designed experiments. M.L.M.-F., D.T.C., S.L., X.L., and L.W. supervised the research. H.Q., M.C.T., and L.W. built the microscope, performed imaging, and data analysis. H.Q., M.C.T., S.K.R., and L.W. prepared fluorescent bead samples, cell samples and STORM imaging buffer. G.L. and R.S. performed the FEA. H.Q., M.C.T., and L.W. wrote the manuscript with input from all authors.

## Competing interests

The authors declare the following competing interests: Shugang Liu is the CEO of RayCage (Zhenjiang) Photoelectric Technology Co., Ltd. Additionally, Shugang Liu, together with Lin Wang, holds more than a dozen of patents related to the optomechanical components designed for the ROCS discussed in this paper. These relationships and intellectual property holdings may be relevant to the subject matter of this paper. The authors declare that these interests have not influenced the results or conclusions presented in this work.
