## [Transparent Peer Review file · Communications Engineering]

Reinforced optical cage systems enable drift-free single-molecule localization microscopy

Corresponding Author: Professor Lin Wang

Version 0:

Reviewer comments:

Reviewer #1

(Remarks to the Author)

Qiu et al. present Reinforced Optical Cage Systems (ROCS), a novel mechanical framework for single-molecule localization microscopy (SMLM) that passively minimizes sample drift by integrating perforated optomechanical components with tungsten-steel reinforcing rods. Finite-element vibration modal analysis predicts elimination of low-frequency rigid-body modes and high natural frequencies (>800 Hz) for the reinforced mounts. Experimentally, a ROCS-based SMLM microscope achieves mean lateral drift of 7.3 ± 1.1 nm (x) and 6.6 ± 1.2 nm (y) over 2 h, a ~ 20 – $40\times$ improvement over a commercial system. In ROCS-STORM, microtubule and nuclear-pore STORM images exhibit ≤ 17 nm drift over 15 min, which is small relative to localization precisions (13–17 nm), rendering post-acquisition drift correction unnecessary. This work offers a low-maintenance, hardware-free route to drift-free SMLM. I would recommend the manuscript to be accepted once the following concerns are addressed:

Major Comments

1. In the Introduction, the authors state:

“Combined with the high laser power (> 1 kW/cm²) used in STORM, samples often drift from their initial locations along lateral (x and y) and axial (z) directions due to the mechanical instability of microscopes and laser-induced heating, both of which degrade the resolution and fidelity of super-resolution images.”

I am not aware of laser-induced drift being a significant issue in SMLM, nor could I find supporting evidence in the cited literature. Please clarify or provide references demonstrating that laser heating at these power densities causes measurable sample drift in SMLM experiments.

2. Widefield drift was measured over five 2 h runs on beads. However, STORM drift results appear to come from a single 15 min acquisition on each sample type. For robust claims of “drift-free” SMLM, please perform multiple independent STORM runs ($n \geq 3$) to establish mean \pm SD and rule out variability due to environmental fluctuations.

The commercial Zeiss AxioObserver Z1 shows significant drift over 2 h (especially in y: 125.8 ± 67.1 nm). Were the ROCS and Zeiss systems operated under identical conditions? For example, was the Zeiss placed on an active optical table? Were temperature and acquisition settings matched? It would strengthen your comparison to include $n > 3$ measurements for both systems under the same conditions.

3. ROCS’s design resembles other commercial cage systems (e.g., Thorlabs). Potential users—particularly DIY microscopists without access to a commercial SMLM platform—would benefit from a direct comparison between ROCS and a standard commercial cage system. Demonstrating ROCS’s performance relative to, say, a Thorlabs cage setup would clarify whether ROCS offers superior stability, ease of assembly, or cost-effectiveness.

4. The manuscript focuses on lateral drift, but axial (z) stability is also critical in SMLM. Did you quantify z-axis drift in the ROCS system, and how does it compare to the commercial microscope? Please report any z-axis measurements or explain why axial drift was not evaluated.

5. You mention a 20 min warm-up and use of a piezo-based autofocus for axial stability. To isolate ROCS’s mechanical contribution to overall drift, please compare lateral drift with and without autofocus engaged. This will help readers understand how much of the improvement comes from the cage reinforcement versus active software/hardware control.

Minor Comments

- Line 121: “optical case systems” should read “optical cage systems.”
- Table S1: Lists parts but omits unit costs. Including approximate pricing would help readers assess feasibility.

Reviewer #2

(Remarks to the Author)

In this paper, Qiu et al. introduced Reinforced Optical Cage Systems (ROCS), a mechanically engineered framework designed to minimize sample drift at the hardware level in single molecule localization microscopy. The authors constructed a drift-free microscope using ROCS, validated its mechanical stability through finite element analysis and experimental measurements, and demonstrated its application in STORM imaging of cellular structures.

The manuscript is well written and presents a compelling case for the hardware-based drift minimization. However, given that Communications Engineering is a relatively new journal, I am uncertain whether this level of engineering innovation fully meets its standard. That said, the experimental data convincingly support the authors' claims.

Comment:

In the discussion of Fig. 5, the authors mention "The increase can be attributed to both the decreased localization precision ... and the additional sample drift caused by thermal expansion due to the higher power density of laser illumination during STORM imaging." However, I would caution against including localization precision as a major contributor to the measured drift. Even though the localization precision of a single fluorophore is 17.5 nm, the drift trajectory is determined by averaging over many molecules. For example, averaging across 100 localizations would reduce the uncertainty in the drift estimate to below 2 nm. Thus, the increased apparent drift is more plausibly attributed to thermal effects alone.

Reviewer #4

(Remarks to the Author)

Review of Qiu et al.

This study addresses the mechanical instability inherent in conventional optical cage systems, which restricts their use in high-precision applications. The researchers developed a Reinforced Optical Cage System (ROCS), inspired by civil engineering, using rigid tungsten steel rods and specially designed perforated components to significantly increase mechanical stability. Finite Element Analysis showed that ROCS eliminates low-frequency rigid-body modes and provides widely separated, higher natural frequencies, compared to conventional cage systems. This indicates much greater resistance to vibrations and mechanical drift. A microscope built on ROCS was tested using fluorescence microscopy of nanometer-sized fluorescent beads. Over two hours, it showed an average drift of about 7 nm along each axis, which was about 20-40 times lower than that of a commercial microscope under identical conditions. Use and performance of the ROCS microscope was shown by STORM Imaging, where final image quality and resolution did not benefit from additional drift correction, demonstrating negligible sample drift during imaging.

The presented system looks technically well-designed and relatively easy to construct using the commercially available parts. It appears to be quite robust and reliable. The presented experimental data convincingly show the capabilities of the system.

The manuscript is well written and sound. I found no obvious flaws. By testing to find the parts on the provided URL, I could not find the items under the given item-numbers or names. Also, the link to the data-repository does not contain any data, yet. This should be corrected.

In conclusion, I recommend this manuscript for publication.

Version 1:

Reviewer comments:

Reviewer #1

(Remarks to the Author)

My comments and concerns have been fully addressed and I believe this manuscript is suitable for publication.

Reviewer #2

(Remarks to the Author)

The authors have addressed my previous concerns, and I have no further comments.

Reviewer #4

(Remarks to the Author)

The authors have responded to the reviewers questions and remarks in sufficient detail and clarity. In my view, the manuscript can be published in the revised version.

Response to Reviewer Comments

Dear Reviewers:

The authors appreciate the time and effort that you dedicated to providing feedback on the manuscript. The authors gratefully appreciate all comments and suggestions, which have helped improve the quality of the manuscript. A point-by-point response to the reviewers' comments and suggestions is provided below in blue.

Response to Reviewer 1

Major Comments

C1. In the Introduction, the authors state:

“Combined with the high laser power ($> 1 \text{ kW/cm}^2$) used in STORM, samples often drift from their initial locations along lateral (x and y) and axial (z) directions due to the mechanical instability of microscopes and laser-induced heating, both of which degrade the resolution and fidelity of super-resolution images.”

I am not aware of laser-induced drift being a significant issue in SMLM, nor could I find supporting evidence in the cited literature. Please clarify or provide references demonstrating that laser heating at these power densities causes measurable sample drift in SMLM experiments.

R1. Thank you for raising the question. We agree that the thermal expansion from high power laser illumination does not significantly contribute to overall sample drift in STORM.

In our study, the thermodynamic simulation results (Methods: Finite Element Analysis, and Supplementary Fig. S26) indicated a maximum thermal expansion of 6.6 nm laterally and 14 nm axially on a coverslip when the temperature increased by 1°C due to laser radiation. Furthermore, our experimental study demonstrated a mean cumulative lateral drift of approximately 5 nm over 2 hours in low-light wide-field microscopy, and 11-16 nm over 15 minutes in the same optical setup, with the main difference being the power density at the sample plane in these two modes. These results indicate that laser heating is a minor contributing factor to sample drift in STORM.

In citation 6 (Liu, Sheng, Philipp Hoess, and Jonas Ries. "Super-resolution microscopy for structural cell biology." *Annual review of biophysics* 51.1 (2022): 301-326.), it was noted that 'mechanical drift exists for all SMLM systems and is often caused by small temperature changes', with the sources of temperature changes attributed to temperature fluctuations and air flow rather than laser-induced heating. Therefore, we have replaced the wording 'laser-induced heating' with 'temperature changes' in the revise manuscript (Main text, page 2, line 51).

C2. Widefield drift was measured over five 2 h runs on beads. However, STORM drift results appear to come from a single 15 min acquisition on each sample type. For robust claims of “drift-free” SMLM, please perform multiple independent STORM runs ($n \geq 3$) to establish mean \pm SD and rule out variability due to environmental fluctuations.

The commercial Zeiss AxioObserver Z1 shows significant drift over 2 h (especially in y: 125.8 ± 67.1 nm). Were the ROCS and Zeiss systems operated under identical conditions? For example, was the Zeiss placed on an active optical table? Were temperature and acquisition settings matched? It would strengthen your comparison to include $n > 3$ measurements for both systems under the same conditions.

R2. Thank you for the suggestion. We agree that reporting mean values of multiple drift measurements in SMLM would strengthen the credibility of this work. Therefore, we have included the results of 3 independent SMLM experiments in the revised manuscript. We present SMLM of α -Tubulin-AF647 in the microtubule network of fixed HeLa cells and Nup96-SNAP-AF647 in fixed U2OS cells in both the ROCS-STORM system (Main text, Fig. 6,7; Supplementary, Fig. S16,17,21,22) and an off-the-shelf microscope (Supplementary, Fig. S18, 19, 20, 23,24,25).

In the ROCS-STORM, the mean drift was 14.4 ± 8.2 nm along the x-axis and 7.3 ± 6.5 nm along the y-axis in HeLa cell imaging (Main text, page 13, line 345), and 6.8 ± 5.2 nm along the x-axis and 8.2 ± 5.2 nm along the y-axis in U2OS cells imaging (Main text, page 16, line 400). In the off-the-shelf microscope, the mean drift was 143.6 ± 124.0 nm along the x-axis and 93.7 ± 60.0 nm along the y-axis in HeLa cell imaging (Main text, page 13, line 362), and 89.3 ± 62.0 nm along the x-axis and 82.0 ± 59.6 nm along the y-axis in U2OS cells imaging (Main text, page 16, line 413). These results further confirm our original conclusion that ROCS-STORM confers minimal sample drift, which is unachievable in standard STORM systems.

We would like to reiterate that the ROCS microscope and the off-the-shelf microscope were operated under identical conditions, as described in the original manuscript (Main text, page 8, line 223; page 10, line 273). The image acquisition settings were the same (Main text, Methods: Imaging fluorescent beads in the ROCS microscope, page 21, line 615; Methods: Imaging fluorescent beads in the ZEISS AxioObserver Z1 microscope, page 22, line 644). Both systems were positioned on optical tables with active-air, self-levelling vibration isolators in the labs with a centrally controlled temperature at 21°C.

C3. ROCS’s design resembles other commercial cage systems (e.g., Thorlabs). Potential users—particularly DIY microscopists without access to a commercial SMLM platform—would benefit from a direct comparison between ROCS and a

standard commercial cage system. Demonstrating ROCS's performance relative to, say, a Thorlabs cage setup would clarify whether ROCS offers superior stability, ease of assembly, or cost-effectiveness.

R3. Thank you for the suggestion. To build another STORM system using conventional optical cage system, e.g. those from Thorlabs, is beyond the scope of this work, as it would require substantial resources and staff time. We address your questions in the following aspects:

1. Stability

In previously reported SMLM setups (1. Ma, Hongqiang, et al. "A simple and cost-effective setup for super-resolution localization microscopy." *Scientific reports* 7.1 (2017): 1542.; 2. Alsamsam, Mohammad Nour, et al. "The miEye: Bench-top super-resolution microscope with cost-effective equipment." *HardwareX* 12 (2022): e00368.), conventional optical cage systems from Thorlabs were deployed. The reported lateral sample drifts reported were 125 nm within 7 minutes and 105 nm within 15 minutes in the former and latter studies, respectively. These drift values are similar to those measured in the off-the-shelf microscope in our study. In conclusion, the mechanical stability of our ROCS outperforms that of conventional optical cage systems.

We added the following comments in the Discussion in the revised manuscript (Main text, page 17, line 437):

'As a comparison, some bench-top STORM systems constructed with conventional optical cage systems^{38,39} [1. Ma, Hongqiang, et al. "A simple and cost-effective setup for super-resolution localization microscopy." *Scientific reports* 7.1 (2017): 1542.; 2. Alsamsam, Mohammad Nour, et al. "The miEye: Bench-top super-resolution microscope with cost-effective equipment." *HardwareX* 12 (2022): e00368.] exhibited sample drift that was comparable to the drift measured in the off-the-shelf microscope of this study, demonstrating the limited mechanical stability of conventional optical cage systems-based assemblies in contrast to the superior stability demonstrated by the ROCS.'

2. Ease of assembly

As stated in the original manuscript (Main text, page3, line 93):

'Evolving from innovative mechanical system designs, ROCS integrate each optomechanical component into a reinforced structure interconnected by tungsten steel rods. A new standardized series of perforated cage components enables the formation of interchangeable ROCS modules.'

ROCS offer improved ease of construction due to the complete freedom to translate perforated cage components along construction rods, whereas in conventional optical cage systems, components are fixed to rods using screws.

We added the following statement to the revised manuscript (Main text, page17, line 446):

'ROCS offer enhanced ease of assembly over conventional optical cage systems through an integrated reinforced framework and modular, perforated cage components. In contrast to conventional optical cage systems, where opto-mechanical components are rigidly fixed to stainless steel rods using screws, ROCS allow unrestricted translation of opto-mechanical components along tungsten steel rods. This flexibility reduces assembly time and facilitates rapid reconfiguration of optical setups.'

3. Cost - effectiveness

We calculated that the total cost of our ROCS-STORM system was about US\$78,000. In a previous low-cost STORM system study [Alsamsam, Mohammad Nour, et al. "The miEye: Bench-top super-resolution microscope with cost-effective equipment." *HardwareX* 12 (2022): e00368.], the reported cost was €50,000, which is comparable to our ROCS-STORM system. A commercially available STORM system would cost more than US\$200,000. Therefore, we conclude that ROCS-based STORM systems offer strong cost - effectiveness.

We added the following comments in the Discussion in the revised manuscript (Main text, page 17, line 443):

'The total cost of the ROCS-STORM system was approximately US\$78,000 (see Table S1), which is comparable to that of other low-cost bench-top STORM system constructed with conventional optical cage systems³⁹ [Alsamsam, Mohammad Nour, et al. "The miEye: Bench-top super-resolution microscope with cost-effective equipment." *HardwareX* 12 (2022): e00368.], and substantially lower than the cost of commercially available instruments.'

C4. The manuscript focuses on lateral drift, but axial (z) stability is also critical in SMLM. Did you quantify z-axis drift in the ROCS system, and how does it compare to the commercial microscope? Please report any z-axis measurements or explain why axial drift was not evaluated.

R3. We agree with you that axial drift is critical in SMLM, therefore we carried out time-lapse measurements of sub-diffraction-limit fluorescent beads while disabling the autofocus function of the piezoelectric stage mounted beneath the objective. We found that the mean PSF size change over five experimental iterations was 2.0 ± 5.6 nm over 2 hours, which was well within the measurement uncertainty.

We also implemented similar measurements with the autofocus function enabled, which reliably maintained the focal plane in axial position during data acquisition. The PSF size change in this case was 0.7 ± 2.1 nm over 2 hours, consistent with our previous findings, given the comparable changes in PSF size over the same time scale. These results demonstrate that ROCS is drift-free not only in the lateral direction but also in the axial direction.

In contrast, the PSF size change was 148.2 ± 54.3 nm in just 5 minutes in an off-the-shelf microscope without any active focal-plane stabilization.

In the revised manuscript, we have reported these findings in Fig 4, S4, S5, S6, S7, with detailed descriptions provided in the main text (page 8, line 204-229).

C5. You mention a 20 min warm-up and use of a piezo-based autofocus for axial stability. To isolate ROCS's mechanical contribution to overall drift, please compare lateral drift with and without autofocus engaged. This will help readers understand how much of the improvement comes from the cage reinforcement versus active software/hardware control.

R5. Thank you for the suggestion. We would like to emphasise that it is unnecessary to use a piezoelectric stage for automatic axial stabilization, as we demonstrated in our previous response 4. However, we agree it is valuable to investigate whether autofocus functions contribute to lateral drift reduction.

We carried out time-lapse measurement of sub-diffraction-limit fluorescent beads with and without autofocus function. The results showed that the drift was 2.5 ± 2.0 nm along the x-axis and 4.0 ± 1.6 nm along the y-axis when the autofocus was disabled, while the values were 4.5 ± 2.0 nm and 5.2 ± 1.7 nm along the x-axis and y-axis, respectively, when the autofocus was enabled. These findings demonstrate that autofocus functions do not contribute to lateral drift reduction, despite their proven capability in eliminating axial drift.

The results have been reported in the main text (page 10, line 253-255, 262-264) and in the Fig. 5, S8, S9, S10, S11, S12.

Minor Comments

C1. Line 121: "optical case systems" should read "optical cage systems."

R1. Thank you for finding this typo. We have corrected it in the revised manuscript (Main text, page 4, line 121).

C2. Table S1: Lists parts but omits unit costs. Including approximate pricing would help readers assess feasibility.

R2. Thank you for the suggestion. We have added the pricing information in the table (Supplementary, page 34, line 290).

Response to Reviewer 2

Major Comments

C1. In the discussion of Fig. 5, the authors mention "The increase can be attributed to both the decreased localization precision ... and the additional sample drift caused by thermal expansion due to the higher power density of laser illumination during STORM imaging." However, I would caution against including localization precision as a major contributor to the measured drift. Even though the localization precision of a single fluorophore is 17.5 nm, the drift trajectory is determined by averaging over many molecules. For example, averaging across 100 localizations would reduce the uncertainty in the drift estimate to below 2 nm. Thus, the increased apparent drift is more plausibly attributed to thermal effects alone.

R1. Thank you for making this plausible point. We think the uncertainty Δ of the mean position from N localizations to be:

$$\Delta = \frac{\sigma}{\sqrt{N}}$$

Where σ is the precision of each localization. When σ is 20 nm and N is 100, the uncertainty is indeed 2 nm. However, in the redundant cross-correlation (RCC)-based drift correction, the estimated drift is calculated between two averaged positions, i.e. two time bins. The uncertainty Δ of a difference of two means is:

$$\Delta = \sqrt{2} \cdot \frac{\sigma}{\sqrt{N}}$$

Therefore, the uncertainty of the estimated drift will be 2.8 nm. Nevertheless, we agree that poorer localization precision from single molecules does not contribute to increased drift measurement results. We have deleted the description stating that decreased localization precision causes increased drift measurement results (Main text, page 18, line 483-486).

Response to Reviewer 4

This study addresses the mechanical instability inherent in conventional optical cage systems, which restricts their use in high-precision applications. The researchers developed a Reinforced Optical Cage System (ROCS), inspired by civil engineering, using rigid tungsten steel rods and specially designed perforated components to significantly increase mechanical stability. Finite Element Analysis showed that ROCS eliminates low-frequency rigid-body modes and provides widely separated, higher natural frequencies, compared to conventional cage systems. This indicates much greater resistance to vibrations and mechanical drift. A microscope built on ROCS was tested using fluorescence microscopy of nanometer-sized fluorescent

beads. Over two hours, it showed an average drift of about 7 nm along each axis, which was about 20-40 times lower than that of a commercial microscope under identical conditions. Use and performance of the ROCS microscope was shown by STORM Imaging, where final image quality and resolution did not benefit from additional drift correction, demonstrating negligible sample drift during imaging.

The presented system looks technically well-designed and relatively easy to construct using the commercially available parts. It appears to be quite robust and reliable. The presented experimental data convincingly show the capabilities of the system.

The manuscript is well written and sound. I found no obvious flaws. By testing to find the parts on the provided URL, I could not find the items under the given item-numbers or names. Also, the link to the data-repository does not contain any data, yet. This should be corrected.

In conclusion, I recommend this manuscript for publication.

Response: We are grateful for your thorough and supportive evaluation of our manuscript.

Regarding the difficulty in locating parts via the provided weblink, we have revised our parts list in Table S1 by adding the hyperlinks to the corresponding Raycege webpages in the 'Part Number' column. This allows the webpages containing the parts information to be accessed directly by clicking the relevant Part Number.

We apologise for the inconvenience caused by a typo in the data repository link. In the revised manuscript, the correct repository information has been updated (Main text, page 24, line 745-747).